## Research Article

posttraumatic stress symptoms; suicidal ideation; mindfulness; spirituality; Palestine

**Corresponding author:**
Fayez Mahamid;
Email: mahamid@najah.edu

# The association between posttraumatic stress symptoms and suicidal ideations among Palestinians: The mediating role of spirituality, social support and mindfulness

Fayez Mahamid[1] , Dana Bdier[1,2], Samah Jabr[3] and Zaynab Hinnawi[1]

[1]Psychology and Counseling Department, An-Najah National University, Nablus, Palestine; [2]University of Milano-Bicocca, Milan, Italy and [3]Community Mental Health Center, Palestinian Ministry of Health, Ramallah, Palestine

## Abstract

The current study aimed to test whether social support, spirituality and mindfulness mediated the association between posttraumatic stress symptoms and suicidal ideation among Palestinians. The study sample consisted of 520 Palestinian adults selected using online tools. The findings of the study revealed that posttraumatic stress symptoms positively correlated with suicidal ideations ($r = .32$, $p < .01$), and negatively correlated with social support ($r = -.34$, $p < .01$), spirituality ($r = -.16$, $p < .05$) and mindfulness ($r = -.72$, $p < .01$), whereas suicidal ideations negatively correlated with social support ($r = -.46$, $p < .01$), spirituality ($r = -.27$, $p < .01$) and mindfulness ($r = -.72$, $p < .01$). Results of the path analysis showed that social support, spirituality and mindfulness mediated the association between posttraumatic stress symptoms and suicidal ideations. Developing intervention programs to enhance social support, mindfulness and spirituality to mitigate the effect of ongoing traumatic experiences among Palestinians is crucial to reducing suicidal ideations. Further studies are also recommended to investigate the role of potentially protective factors that may reduce suicidal ideations and appropriate techniques to deal effectively with traumatic and stressful events.

## Impact statement

Suicide is difficult to address in Palestine because it is considered as a crime and a sin. Several suicide cases are reported annually in Palestine, but it is important to look at the numbers of other violent deaths, to make sense of available data on suicide. Suicide cases in Palestine continue to record a remarkable annual increase, but we lack a comprehensive verified national data collection system of suicide and it is expected that real numbers of attempted/suicide in Palestine are higher because not all suicide or attempted suicide cases are reported. Testing the role of some protective factors against suicide such as social support, spirituality and mindfulness is crucial to reduce suicidal ideations and attempts, especially in a context characterized by a high level of stressors and ongoing traumatic experiences.

## Theoretical background

Living in fragile areas affected by occupation can have catastrophic and debilitating effects on several different aspects of individuals' life (Singhal, 2019). In occupied countries, such as Palestine, people are continually subjected to: high rates of poverty and unemployment, dependence on the economy, infringements against personal justice and equality, fragmentations of territories, restrictions on buildings, invasions and demolitions of homes. These actions are compounded with pressures brought on by culture, an unsecure future, reduced social avenues, few recreational facilities, as well as poor health and mental health care services, and a ban on social networks that make it all the more evident in noticing such negatives effect on the population (Sousa et al., 2014; Barber et al., 2016; Berte et al., 2019; Mahamid and Bdier, 2020; Mahamid and Veronese, 2021; Veronese et al., 2021).

Experiencing such challenging and hard life conditions were found to negatively affect the mental health of Palestinians, as they were found to suffer from depression, anxiety and stress due to the actions of the Israeli military occupation (Marie and SaadAdeen, 2021; Bdier et al., 2022). In addition, living under such constrictions were found to negatively affect the quality of life and well-being of Palestinians, as one-third reported low levels of well-being (Mataria et al., 2009; Harsha et al., 2016), which was found to correlate positively with feelings of hopelessness and

depressive symptoms among Palestinians in the West Bank (Mahamid et al., 2022; Veronese et al., 2022).

The feelings of insecurity, poor socioeconomic status and the continued vulnerability to violence brought on by politics were closely linked with symptoms of depression, high levels of depression and suicidal tendencies among Palestinian adults (McNeely et al., 2014; Wagner et al., 2020; Hamdan and Hallaq, 2021). A recent comparative study found that populations with some of the highest percentages of suicidal tendencies among the 22 countries surrounding the Mediterranean Sea were found to be in Palestine, along with Cyprus, Greece and Slovenia (Eskin, 2020). According to the Research Department of Palestinian Police, more Palestinians are attempting and committing suicide as the reported suicidal cases in the West Bank of Palestine increased by 14% in 2018 compared with 2017 (Abu Ghoush, 2022).

Spirituality, on the other hand, is deemed to be a protective factor against suicidal ideations that stem from stressful life events, and can be defined as the living reality of religion as witnessed by individuals who follow and practice specific traditions. It is comprised of four important principles: (1) brings feeling of purpose beyond oneself, and the belief in a higher power in looking for guidance (2) a clearer interpretation of life, (3) self-awareness and (4) self-reflection as part of self-healing (Nelson, 2009).

Mental health may also by protected by spirituality, as it is said to enhance the influence of positive emotions such as hope, forgiveness, self-esteem and love through the practice of spiritual activities (Salgado, 2014). Furthermore, spiritual and religious beliefs and practices may help people to better cope with stressful life circumstances, give meaning and hope, as it surrounds depressed individuals with a supportive community (Bonelli et al., 2012).

Protection against suicidal ideations has also been associated with spirituality (Rasic et al., 2011). In a study that investigated the role of religiousness in moderating the impact of stressors on depressive symptoms, the results showed that various dimensions of religiousness buffered the impact of life events on depressive symptoms (Lorenz et al., 2019). Moreover, a study investigating the effects of spirituality on a sample of Indian students attending a religious university, found that it had a positive impact on the well-being of the participates, which included the potential to protect against suicidal behaviors (Wagani and Colucci, 2018).

In addition to spirituality, social support carries with it a sense of appreciation, love and a belonging to a specific community that humans, as social creatures, desperately need to help them cope with stressors in life, and therefore can be seen as one of the most influential factors in people's life (Agbaria and Bdier, 2020; Mahamid et al., 2023).

Griffith (2012) found that social support has a mediating effect on the outcomes of wartime experiences as they relate to posttraumatic stress disorder (PTSD), negative moods and suicidality among army soldiers. Eskin et al. (2021) investigated suicidality in a cross-national study, sampling college students from 11 predominantly low- to middle-income majority Muslim countries. The results revealed that suicide acceptance and negative life-events were associated with suicidal ideation, whereas perceived social support was negatively related to suicidal ideation. Moreover, Kleiman et al. (2014) explored the role of social support and positive events as protective factors in suicide. Such results signified that social support in connection with positive events protected against suicide ideation, acted as intermediaries in the relationship between negative events and suicide ideation; and mediated the symbiotic relationship between negative events and suicide ideation.

Another expected protective factor is mindfulness, a state in which individuals alter their cognitive schema from threat appraisals to positive reappraisals that rely on metacognition, a naturalistic state wherein consciousness transforms its content to comfort the dynamics of its own processes. In other words, the individual's mental health will be empowered and enhanced by such new and positive appraisals as they are considered to be essential workings of coping through understanding meaning, which in turn allows individuals to successfully adapt to the stresses in their lives (Garland, 2007; Garland et al., 2009).

According to literature, teaching mindfulness techniques are expected to be important strategies toward the prevention of suicide, especially among young adults experiencing stress induced and depressive symptoms (Anastasiades et al., 2017). In one study, Song and Bae (2022) examined the moderating role of mindfulness on the relationship between daily life stresses and suicidality among Korean college students. Results revealed that mindfulness moderated the effect of daily life stresses on suicidal ideation. Furthermore, Deng et al. (2014) explored the relationship between a wandering mind and depression and mindfulness, and it was found that depression was negatively related to dispositional mindfulness.

### Current study

Palestinians are experiencing several stressors on a daily basis due to the ongoing Israeli occupation (Sousa et al., 2014; Barber et al., 2016; Berte et al., 2019; Mahamid and Bdier, 2020; Mahamid and Veronese, 2021; Veronese et al., 2021), where suicide is considered to be on the rise in the Palestinian context (Eskin, 2020; Abu Ghoush, 2022). According to previous literature (Garland et al., 2009; Rasic et al., 2011; Griffith, 2012; Deng et al., 2014; Kleiman et al., 2014; Wagani and Colucci, 2018; Eskin et al., 2021; Song and Bae, 2022), study hypotheses were defined as: *(1) Posttraumatic stress symptoms would be positively correlated with suicidal ideations. (2) Spirituality, social support and mindfulness would mediate the correlation between posttraumatic stress symptoms and suicidal ideations among Palestinians.*

### Methodology

#### *Participants and procedures*

The aims of this study, along with the procedures, were presented through online forums in October of 2022. Participants were recruited through online advertisements, e-mail campaigns and social media. Interested participants sent an e-mail indicating their willingness to participate in response to study recommendations. Each participant was then sent an e-mail briefly explaining the subject of the study along with its purpose, noting all ethical issues of confidentiality and voluntary participation. Participants were instructed to reply with their informed consent upon accepting the conditions outlined in the e-mail.

In order to establish the sample size for this study, the Raosoft software sample size calculator was used with 95% CI and 5% margin of error, whereby the recommended sample was found to be 520. The 520 Palestinian adults consisted of 219 men and 301 women. About 51.4% of participants were from cities and 41.3% were from Palestinian villages, 7.3% of participants were from Palestinian internally displaced camps. About 18.7% participants had a master degree, and 56.3% of participants had a bachelor's degree, whereas the remaining participants were without an academic degree. Participants' age ranged from 18 to 53 years old ($M = 34.2$, SD $= 14.21$).

For inclusion in the study, participants are required to be native Arab speakers, Palestinians and living in the West Bank of the

occupied Palestinian territories (oPt). The ethical guidelines of the American Psychological Association (APA, 2010) and the Declaration of Helsinki (2013) guided the way in which this research was conducted, and approval was given by the An-Najah Institutional Review Board (IRB).

### Measures

Proper standard methodological procedures were followed in developing questionnaires for this research (Hambleton et al., 2004). First, an independent expert in both Arabic and English language editing translated all items to Arabic and back-translated them into English. Second, a panel of 10 Arab professionals identified as experts in psychology, counseling and social work verified the clarity and relevance of the questions and translation. Third, 80 participants (known as the validation sample) test piloted the translated version, which was then later modified for clarity.

*The Impact of the Event Scale (IES-R)*: The IES-R is a self-report measure consisting of 22 items designed to test posttraumatic stress symptoms in response to several traumatic and stressful events. The IES-R scale including three sub-scales to represent the main symptoms of traumatic events: Hyperarousal, avoidance and numbing, and instruction. The hyperarousal subscale including several items to represent anger, irritability and difficult concentration related to traumatic experiences, while avoidance and numbing scale including several items on avoidance of people, places and things related to traumatic experiences. Finally, the intrusion subscale including several items designed to test nightmares, distressing thoughts and extreme feelings related to traumatic experiences (Weiss and Marmar, 1997).

*Berlin Social Support Scales (BSSS)*: The BSSS is a self-report including six subscales designed to measure behavioral, social and cognitive aspects related to social support, including actually received support (which related to the actual support received from individuals), perceived available support (which related to the availability of support provided by others), need for support (which related to the need of social support in different situations), provided support (this subscale completed by individuals who provide support to the respondent) and protective-buffering subscale (this subscale related to protecting family members, close friends and colleagues who offer social support to the person during crisis and difficult situations; Schulz and Schwarzer, 2013).

*The Scale for Suicide Ideation (SSI)*: The SSI is a self-report scale consists of 19 items designed to test suicidal ideations through three sub-scales, specific plans for suicide, active suicidal desire and passive suicidal desire (Eskin, 2020). SSI is a three-point Likert scale ranging from 0 to 2, the greater degree on the scale indicates the greater severity of suicidal ideation among respondents. The SSI indicated a high level of internal consistency in evaluating suicidal ideations in the Palestinian context (α = .91).

*Daily Spiritual Experiences Scale (DSES-C17)*: The DSES-C17 is a 17 items self-report designed to evaluate the degree of spiritual experiences among respondents (e.g., "I feel deep inner peace or harmony"; "I find strength in my religion or spirituality"). The items of DSES-C17 were rated on a five-point Likert scale, ranging from 1 (never) to 5 (always). The greater degree on the scale indicates high level of spirituality among participants. The DSES-C17 indicated a high-internal consistency in assessing spirituality well-among Palestinians (α = .91).

*The Mindful Attention Awareness Scale (MAAS)*: The MAAS is a self-report scale which was designed to measure mindfulness in

several daily experiences (Park et al., 2013). All items of the scale are negatively worded (e.g., "I find it difficult to stay focused on what's happening in the present"). Items are rated on a six-point Likert scale ranging from 1 (always) to 6 (never). The greater degree on the scale indicates a high level of mindfulness among participants. The MAAS indicated a high-internal consistency in assessing mindfulness among Palestinians (α = .93).

### Data analysis

We used structural equation modeling (SEM) to test our conceptual model, where posttraumatic stress symptoms considered as a predictor variable. While, suicidal ideation identified as an outcome variable. Finally, spirituality, social support and mindfulness were operated as mediating variables. The model showed goodness of fit, RMSEA = .057, SRMR = .048 and CFI = .88. Moreover, descriptive statistics were used to explore the characteristics of our study variables; in addition, Pearson's correlation coefficient was used to test the correlation between posttraumatic stress symptoms, social support, suicidal ideations, spirituality and mindfulness. To test our conceptual model (Figure 1), AMOS 25 software was used for data analysis.

### Findings

The study calculated descriptive statistics for posttraumatic stress symptoms, suicidal ideation, social support, spirituality and mindfulness as indicated in Table 1. Results indicated that participants recorded high scores on social support and spirituality, and moderate scores on posttraumatic stress symptoms and mindfulness, and within low scores on suicidal ideations. Moreover, measures of our study indicated a high level of reliability ranging from .87 (mindfulness) to .93 (spirituality).

Results of the correlational analysis in Table 2 showed that posttraumatic stress symptoms positively correlated with suicidal ideations ($r = .32$, $p < .01$), and negatively correlated with social support ($r = -.34$, $p < .01$), spirituality ($r = -.16$, $p < .05$) and mindfulness ($r = -.72$, $p < .01$), whereas suicidal ideations negatively correlated with social support ($r = -.46$, $p < .01$), spirituality ($r = -.27$, $p < .01$) and mindfulness ($r = -.72$, $p < .01$). Social support positively correlated with spirituality ($r = .48$, $p < .01$), and mindfulness ($r = .26$, $p < .01$). Finally, spirituality positively correlated with mindfulness ($r = .17$, $p < .05$).

### Structural equation modeling (SEM)

SEM results are shown in Figure 2 along with the hypothesized model in Figure 1 with posttraumatic stress symptoms operated as a predictor, social support, spirituality and mindfulness as mediating variables, and suicidal ideation as an outcome variables which were then tested across the sample ($n = 520$). The study showed that social support, spirituality and mindfulness mediated the correlation between posttraumatic stress symptoms and suicidal ideations. Finally, the paths of our model were significant with good fit indicators ($\chi^2_{(3)} = 40.20$; $p = .001$; GFI = .94; AGFI = .95; RMSEA = .05; NFI = .956; CFI = .95).

The results of path analysis showed a positive effect between posttraumatic stress symptoms and suicidal ideations ($\beta_{X,Y} = -.41$; $p < .001$). Moreover, a negative effect was found between posttraumatic stress symptoms and (social support $\beta_{X,Y} = -.26$, $p < .005$;

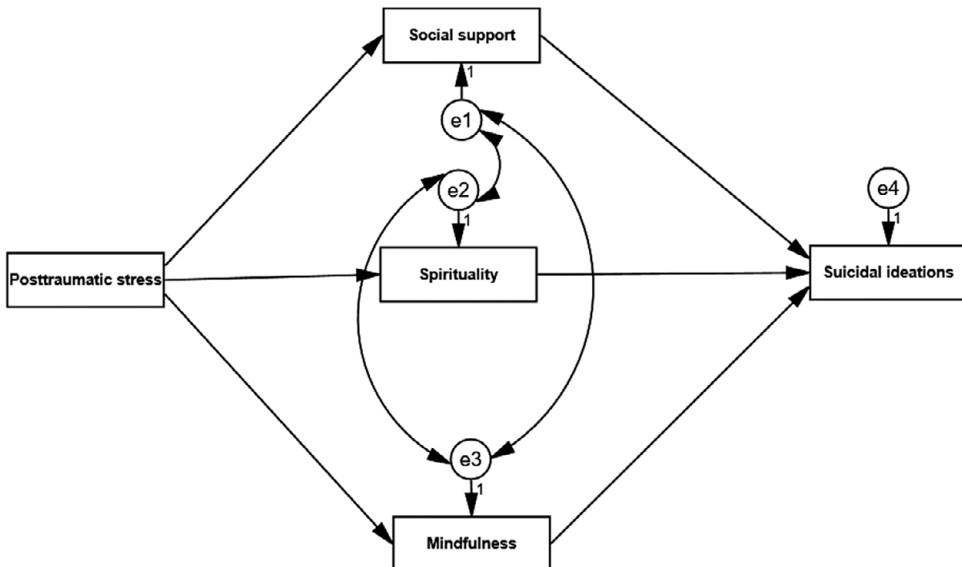

**Figure 1.** The conceptualized effect of posttraumatic stress on suicidal ideation, and the mediating role of social support, spirituality and mindfulness.

**Table 1.** Descriptive statistics for research variables *(N = 520)*

| Variable | M | SD | Min | Max | Range | Skewness | Kurtosis | Cronbach's alpha |
|---|---|---|---|---|---|---|---|---|
| Posttraumatic stress symptoms | 2.44 | .82 | 1.00 | 4.24 | 3.24 | .34 | −.60 | .91 |
| Suicidal ideation | .53 | .22 | .30 | 1.50 | 1.20 | 2.38 | 6.11 | .89 |
| Social support | 3.84 | .57 | 1.67 | 4.44 | 2.78 | −1.47 | 3.10 | .90 |
| Spirituality | 3.67 | .38 | 2.22 | 4.00 | 1.78 | −1.51 | 1.75 | .93 |
| Mindfulness | 3.06 | .80 | .45 | 4.82 | 4.36 | −.36 | .52 | .87 |

**Table 2.** Correlations among study variables *(N = 520)*

| Measures | 1 | 2 | 3 | 4 | 5 |
|---|---|---|---|---|---|
| 1. Posttraumatic stress symptoms | 1 | .32** | −.34** | −.16* | −.72** |
| 2. Suicidal ideation | | 1 | −.46** | −.27* | −.26* |
| 3. Social support | | | 1 | .48** | .26* |
| 4. Spirituality | | | | 1 | .17* |
| 5. Mindfulness | | | | | 1 |

**α is significant at ≤.01.
*α is significant at ≤.05.

mindfulness $\beta_{X,Y} = -.62$, $p < .001$; spirituality $\beta_{X,Y} = -.16$, $p < .005$). In addition, a negative effect was found between social support and suicidal ideations ($\beta_{M,Y} = -.28$; $p < .001$), whereas a negative effect in path analysis was found between mindfulness and suicidal ideation ($\beta_{M,Y} = -.19$; $p < .005$). Furthermore, a negative effect in path analysis emerged between spirituality and suicidal ideations ($\beta_{M,Y} = -.20$; $p < .005$).

Concerning the mediating hypothesis (H2), our model showed a standardized total effect of social support on suicidal ideations ($\beta_{X,M} = -.46$; $p < .001$). However, this effect was composed of a statistically significant indirect effect (via social support $\beta_{X,M,Y} = -.14$; $p < .05$) and a statistically significant direct effect ($\beta_{X,Y,M} = -.32$ $p < .01$). Moreover, the model showed a total effect of spirituality on suicidal ideations ($\beta_{X,M} = -.43$; $p < .001$), with indirect effect (via spirituality $\beta_{X,M,Y} = -.14$; $p < .05$) and a statistically significant direct effect ($\beta_{X,Y,M} = -.29$; $p < .01$). Finally, our model showed a standardized total effect of mindfulness on suicidal ideations ($\beta_{X,M} = -.52$; $p < .001$), with significant indirect effect via mindfulness ($\beta_{X,M,Y} = -.16$; $p < .05$) and significant direct effect ($\beta_{X,M,Y} = -.36$; $p < .01$). These paths mean that the correlation between posttraumatic stress symptoms and suicidal ideations among Palestinians was mediated by social support, spirituality and mindfulness.

## Discussion

This study investigated the relationship between posttraumatic stress symptoms and suicidal ideations among a sample group of

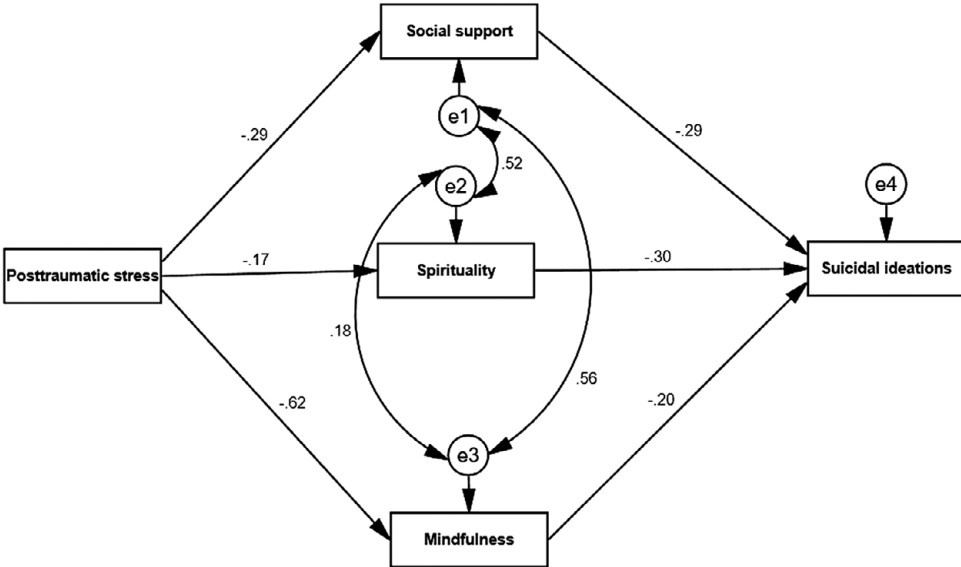

**Figure 2.** Structural equation modeling of posttraumatic stress on suicidal ideation, and the mediating role of social support, spirituality and mindfulness.

Palestinian adults. It also aimed to investigate whether spirituality, social support and mindfulness mediate the association between these variables. The study's findings revealed a significant positive association between posttraumatic stress symptoms and suicidal ideations, and that spirituality, social support and mindfulness mediated the association between posttraumatic stress symptoms and suicidal ideations. The findings of this study are consistent with previous studies also investigating the effect of posttraumatic stress symptoms on suicidal ideations.

A study by Borges et al. (2008) tested the relationship between posttraumatic stress symptoms and suicidal ideation, suicide plans and suicide attempts among Mexican adolescents. The results showed that the prevalence of suicidality was high among respondents with posttraumatic stress symptoms. Dogra et al. (2008) examined the role of posttraumatic stress symptoms in predicting suicidal ideation of college students, and the findings suggested that posttraumatic stress symptoms contribute significantly in suicidal ideation. Sandin et al. (1998) investigated the relationships between negative life events and adolescent suicidal behavior using systematic reviews. The results supported the hypothesis that life events may be a comprising risk factor for adolescent suicidal behavior.

In regards to this study, one possible explanation of the results is that the situation of individuals in the occupied territories of Palestine is fraught with environmental stressors (poverty, military occupation, lack of employment opportunities, cultural pressures, etc.) and few positive social avenues. These constrictions are in part due to the occupation's enforced restrictions on movement between regions, cultural standards of gender separation, and a lack of recreational facilities. In these arduous situations, it may come to be expected that Palestinians may exhibit suicidal ideation in the face of increasing traumatic stressors.

Tendencies toward suicidal ideation and behavior may often develop among people who have experienced traumatic events. Such individuals may include war veterans, victims of interpersonal violence, individuals who suffered from childhood abuse and neglect, survivors of natural disasters, people exposed to torture and collective trauma victims and survivors. Suicide risk may also be elevated in case of secondary (or vicarious) traumatization (Krysinska et al., 2009).

The findings indicated that social support mediated the association between posttraumatic stress symptoms and suicidal ideation. One probable explanation of this result is that social support, which is characterized by a high level of collectivism, may mitigate the effect of traumatic events, especially in the Palestinian context. Social support thrives in collectivist cultures. In this culture, people from birth onward are part of stable and close-knit communities, which allow people to feel protected throughout their lifetime in exchange for unquestioning loyalty.

In addition, numerous studies revealed that social support mediated between traumatic events and suicidal ideations by acting as a buffer. A study by Zheng et al. (2021) was designed to test the relationship between traumatic events and current suicidal ideations among a sample of Chinese female prisoners and examine the mediating role of social support. SEM analyses revealed that social support acted as a mediator between trauma and current suicidal ideations. Wang et al. (2019) explored the interactions between posttraumatic stress symptoms, suicidal ideations and social support among Chinese parents who lost their children. The results indicated that posttraumatic stress symptoms were positively associated with social ideations, whereas social support buffered the effect of posttraumatic stress symptoms on suicidal ideation. Panagioti et al. (2014) examined whether perceived social support reduced the impact of posttraumatic stress symptoms on suicidal behaviors. Results revealed that perceived social support lowered the impact and severity of posttraumatic stress symptoms on suicidal behaviors.

Our findings also revealed that spirituality mediated the association between posttraumatic stress symptoms and suicidal ideations. The current findings can be explained through which Palestinian people practice positive religious coping and spirituality to deal with stressful events, which in turn lead them to strengthen in their relationship with God. This may then lead them to cope in a positive manner toward stressful events, and may help them to deal with the debilitating impacts of traumatic experiences. Spirituality and positive religious coping may give traumatized people the necessary guidance and as well as a sense of support to face life's challenges, particularly during difficult situations (Mahamid and Bdier, 2021). Several studies emphasized the role of spirituality in

mitigating posttraumatic stress symptoms and suicidal ideations among traumatized people, for example, Florez et al. (2017) examined the extent to which spiritual well-being had toward suicidal ideation and posttraumatic stress symptoms. Findings indicated that spiritual well-being mediated the relationship between levels of posttraumatic stress symptoms as well as levels of hopelessness and suicidal ideation in terms of severity. Richardson et al. (2022) investigated the association between spirituality, traumatic stressors and suicidal ideation. Their findings suggested that participants who exhibited low levels of spirituality were found to have higher levels of posttraumatic stress symptoms and suicidal ideations.

The findings of this study showed that mindfulness mediated the association between posttraumatic stress symptoms and suicidal ideation. One possible explanation for this result is that individuals with high level of awareness tend to stay focused and not allow their mind to wander. These individuals may display greater ability to recognize that a thought is a just a thought, which is, in turn, associated with reduced risk of suicide (Stanley et al., 2019).

Non-judgmental individuals, for instance, tend not to judge their inner experiences too harshly. For example, in having negative feelings such as shame and guilt, these individuals tend not to judge having these feelings, and instead may conclude a logical reason as to why they are exhibiting such feelings as understandable responses to circumstances in their lives. Emotions such as shame and guilt are closely tied to posttraumatic stress symptoms and the risk of suicide, and therefore, these individuals may be better suited to combat posttraumatic stress symptoms and suicide (Gallegos et al., 2015).

It is logical and not surprising that higher levels of mindfulness facets attenuate the association between posttraumatic stress symptoms and suicide risk among Palestinians. Several studies have supported the mediating role of mindfulness between posttraumatic stress symptoms and suicidal ideations. Cheng et al. (2018) examined the association between PTSD and suicidality and the mediating role of mindfulness. Results indicated a reversal significant effect of mindfulness on suicidal ideations. Kachadourian et al. (2021) evaluated whether mindful attention to trauma in the present moment mediated the association between the number of lifetime traumas and suicidal ideation. Results of the path analyses revealed that mindfulness fully mediated the association between number of lifetime traumas and suicide ideations.

### *Limitations of the study*

The current study presents several limitations that may offer opportunities for future research. First, this study was carried out during a heightened period of political violence in the West Bank of Palestine. Furthermore, longitudinal studies targeting the responses of Palestinians to different political climates are needed to generalize the findings. Second, the current study only used a cross-sectional quantitative design to investigate the impact posttraumatic stress symptoms had on suicidal ideations among Palestinians. It is recommended that future studies use mixed methods design to collect and interpret data. Third, testing and validation of the study instruments within the Palestinian context were not considered; therefore, testing the psychometric properties of these measures in future studies is needed.

### Conclusion

No previous studies ventured to investigate the relationship between posttraumatic stress symptoms and suicidal ideation in Palestine. The current study, therefore, is the first of its kind designed to investigate the correlation between such factors since Palestinians live in a society characterized by high levels of political violence and ongoing trauma. This study also factored in whether or not mindfulness, social support and spirituality mediated the associated between posttraumatic stress symptoms and suicidal ideations. The results of this study supported the central hypothesis that repeated exposure to posttraumatic stress symptoms may heighten suicidal ideations. The findings also supported the mediating hypothesis that spirituality, social support and mindfulness mediate the correlation between posttraumatic stress symptoms and suicidal ideation. There is also a need to conduct other studies to investigate the role of specific traumatic experiences in predicting traumatic symptoms, and the other possible protective factors that may help reduce suicidal ideation among Palestinians and bring forth appropriate techniques that may assist them to effectively deal with traumatic and stressful events. Developing different intervention programs that focus on mental health and suicide prevention may help to enhance social support, mindfulness and spirituality which are crucial to mitigating the effects of ongoing traumatic experiences among Palestinians and may lead to a reduction against suicidal ideations.

**Open peer review.** To view the open peer review materials for this article, please visit http://doi.org/10.1017/gmh.2023.41.

**Data availability statement.** All data generated or analyzed during this study are included in this published article.

**Author contribution.** F.M. and D.B. prepared the theoretical background and the methodology sections; F.M. collected the data and analyzed it. S.A. and Z.H. prepared the discussion sections. All authors commented on the drafts of the manuscript. All authors read and approved the final manuscript.

**Financial support.** This research did not receive any grant from funding agencies in the public, commercial, or non-for-profit sectors.

**Competing interest.** The authors declare no competing interest exists.

**Ethics standard.** All procedures performed in this study involving human participants were in accordance with the ethical standards of An-Najah National University Institutional Review Board (IRB), the American Psychological Association (APA, 2010) and with the 2013 Helsinki Declaration. Informed consent was obtained from all participants.

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
