## [Reviewer Report]

Dear Editors, 

We are glad to submit our paper entitled The association between posttraumatic stress symptoms and suicidal ideations among Palestinians: The mediating role of spirituality, social support, and mindfulness. This is an important study testing the correlation between posttraumatic stress symptoms and suicidal ideations and whether spirituality, social support, and mindfulness mediate the correlation between the two variables in a context characterized by a high level of stress and ongoing trauma. Hoping the manuscript can be interesting for the readers of the Global Mental Health Journal. All authors declare no conflicts of interest and agree for submitting to the journal.

Thank you for your gentle understanding, 

Yours Sincerely, 

Fayez Mahamid (on behalf of the co-authors).

---

## [Reviewer Report]

The article examines several variables of interest including the relationship between suicidality and trauma symptoms , the function of spirituality as a mediator for mental health symptoms, and could potentially examine the relationship between mindfulness and spirituality.

In the future I would highly suggest the a measure of traumatic experiences be added to better understand both the actual prevalence of traumatic experiences in the general population and the role of a quantified measure of trauma in predicting reaction to trauma (symptomology) and differentials of spiritual methodologies in managing traumatic experiences.

Despite the lack of data on actual trauma experiences the article does a good job of investigating the phenomenon.

Some English language editing is needed but otherwise the article is appropriate for publication in. it’s current state.

---

## [Reviewer Report]

Thank you for the great article. the topic is relevant in the Palestinian context, and it is not researched due to the nature of the topic (suicide). the sample was enough, and the online version is appropriate due to sensitivity. the key words should include post-traumatic stress; please follow APA in reference orders within the text. the theoretical aspect is comprehensive; the analysis is excellent, and the discussion is great. The conclusion can use another paragraph or 2. thank you for the great article.

---

## [Reviewer Report]

Dear Editors,

Thank you for your great efforts and constructive comments concerning our article entitled “The association between posttraumatic stress symptoms and suicidal ideations among Palestinians: The mediating role of spirituality, social support and mindfulness” These comments are valuable and beneficial for improving this article. We have tried our best to modify the manuscript to meet the requirements of your journal. In this revised version, changes to this manuscript within the document were all highlighted using red-coloured text. Point-by-point responses to the editors are listed below. If there are any other modifications we could do, we would like to modify them, and we appreciate your help.

Editor’s comments 

- Please include the abstract in the main text document.

We appreciate the editor’s feedback and have included the abstract in the main document (see p.1)

- Please include an Impact Statement below the abstract (max. 300 words). This must not be a repetition of the abstract but a plain worded summary of the wider impact of the article. 

We included an impact statement (see p.2)

- Submission of graphical abstracts is encouraged for all articles to help promote their impact online. A Graphical Abstract is a single image that summarises the main findings of a paper, allowing readers to quickly gain an overview and understanding of your work. Ideally, the graphical abstract should be created independently of the figures already in the paper, but it could include a (simplified version of) an existing figure or a combination thereof. Graphical abstracts should not be too text-heavy in order to be easily viewable at thumbnail size. If you do not wish to include a graphical abstract please let me know. 

We greatly appreciate your feedback. Actually our manuscript includes SEM diagrams clarifying the main results of our study (see Figure 2, page30). Therefore, we believe there is no need to add a graphical abstract to our manuscript. 

- Please ensure references are correctly formatted. In text citations should follow the author and year style. When an article cited has three or more authors the style ‘Smith et al. 2013’ should be used on all occasions. At the end of the article, references should first be listed alphabetically, with a full title of each article, and the first and last pages. Journal titles should be given in full.

We checked all references in the text and in the list of bibliography.

- Statements of the following are required in the main text document at the end of all articles: ‘Author Contribution Statement’, ‘Financial Support’, ‘Conflict of Interest Statement’, ‘Ethics statement’ (if appropriate), ‘Data Availability Statement’. Please see the author guidelines for further information. 

- Please submit figures as separate files and please ensure all files are submitted in an editable electronic format.

We added the corrections ( see p.19)

Reviewer(s)' Comments to Author:

Reviewer: 1

Comments to the Author

The article examines several variables of interest including the relationship between suicidality and trauma symptoms , the function of spirituality as a mediator for mental health symptoms, and could potentially examine the relationship between mindfulness and spirituality.

We greatly appreciate the reviewer’s positive back concerning our article 

In the future I would highly suggest the a measure of traumatic experiences be added to better understand both the actual prevalence of traumatic experiences in the general population and the role of a quantified measure of trauma in predicting reaction to trauma (symptomology) and differentials of spiritual methodologies in managing traumatic experiences. Despite the lack of data on actual trauma experiences the article does a good job of investigating the phenomenon.

We appreciate the reviewer’s feedback and have added this recommendation to the conclusion section ( see p.18 )

Some English language editing is needed but otherwise the article is appropriate for publication in. it’s current state.

We appreciate the reviewer’s feedback and have done thorough English language editing for our manuscript

Reviewer: 2

Comments to the Author

Thank you for the great article. the topic is relevant in the Palestinian context, and it is not researched due to the nature of the topic (suicide). the sample was enough, and the online version is appropriate due to sensitivity. 

We greatly appreciate the reviewer’s positive feedback concerning our article 

The key words should include post-traumatic stress; please follow APA in reference orders within the text.

We appreciate the reviewer’s feedback and have added posttraumatic stress symptoms to the keywords (see p.1)

The theoretical aspect is comprehensive; the analysis is excellent, and the discussion is great. 

We appreciate the reviewer’s feedback

The conclusion can use another paragraph or 2. thank you for the great article

We appreciate the reviewer’s feedback and have added some thoughts to our conclusion section (see p.18)

With best regards